# Emergence of NDM-1- and OXA-23-Co-Producing *Acinetobacter baumannii* ST1 Isolates from a Burn Unit in Spain

**DOI:** 10.3390/microorganisms13051149

**Published:** 2025-05-16

**Authors:** Elena Hidalgo, Jared Sotelo, María Pérez-Vázquez, Ángela Iniesta, Javier E. Cañada-García, Olga Valiente, Belén Aracil, David M. Arana, Jesús Oteo-Iglesias

**Affiliations:** 1Servicio de Microbiología, Hospital Universitario de Getafe, 28905 Madrid, Spain; 2Laboratorio de Referencia e Investigación en Resistencia a Antibióticos e Infecciones Relacionadas con la Asistencia Sanitaria, Centro Nacional de Microbiología, Instituto de Salud Carlos III, 28222 Madrid, Spain; 3Centro de Investigación Biomédica en Red de Enfermedades Infecciosas (CIBERINFEC), Instituto de Salud Carlos III, 28029 Madrid, Spain

**Keywords:** *Acinetobacter baumannii*, antimicrobial resistance (AMR), NDM-1, OXA-23, ICU infections, whole-genome sequencing (WGS)

## Abstract

The global emergence of carbapenem-resistant *Acinetobacter baumanii* (CRAB) represents a significant public health threat. In the summer of 2022, a polyclonal CRAB outbreak occurred in our hospital, marking the first detection of an NDM-1 plus OXA-23 co-producing *A. baumannii* strain in Spain. The aim of this study was to phenotypically and genotypically characterize the clonal spread of NDM-1 and OXA-23 co-producing *A. baumannii* isolates and to describe the infection control measures implemented to contain the outbreak. Patients with multidrug-resistant *A. baumannii* isolates (July 2022–May 2023) were included in the study. Isolates were identified via MALDI-TOF, and antimicrobial susceptibility was tested using a broth microdilution method (DKMGN SensititreTM panels). Whole-genome sequencing was performed on 24 representative isolates. Phylogenetic analysis was performed using Ridom SeqSphere+ (cgMLST), while sequence typing was performed using ARIBA (Pasteur and Oxford schemes). *A. baumannii* isolates from the affected patients belonged to five different sequence types. The two main STs were ST1Pas/ST231Oxf (NDM-1- and OXA-23-co-producing), which accounted for 58%, and ST136Pas/ST406Oxf (OXA-23-producing), which accounted for 21%. All isolates were resistant to fluoroquinolones, trimethoprim/sulfamethoxazole, aminoglycosides, and carbapenems. In addition, 8% were resistant to colistin and 17% to cefiderocol. Finally, the affected patients were cohorted, and a thorough cleaning of the affected units was carried out. This study documents the clonal spread of an NDM-1- and OXA-23-co-producing *A. baumannii* strain in Spain, linked to a Libyan patient, highlighting the risk of cross-border spread. Although infection control measures successfully contained the outbreak, surveillance is essential as the incidence of CRAB infections is expected to increase.

## 1. Introduction

Antimicrobial resistance is emerging as one of the most important threats to global public health, threatening the effective treatment of an increasing number of bacterial infections in various healthcare settings [1]. Multidrug-resistant *Acinetobacter baumannii* (MDRAB) has been identified as one of the most important pathogens, with high rates of resistance to multiple classes of antibiotics [2]. Recent genomic and phenotypic analyses of *A. baumannii* have identified several virulence factors responsible for its pathogenicity, including porins, capsular polysaccharides, lipopolysaccharides, phospholipases, outer membrane vesicles, metal acquisition systems, and protein secretion systems [3,4]

The rapid emergence of MDRAB has led to a worrying situation worldwide. MDRAB is an opportunist pathogen that, in recent decades, has emerged as one of the main causes of nosocomial infection, mainly affecting patients admitted to intensive care units (ICUs) and burn units.

A multinational study of ICUs revealed that the prevalence of MDRAB was 14.8% in Africa, 5.6% in Western Europe, 3.7% in North America, 13.8% in Central and South America, 17.1% in Eastern Europe, 4.4% in Oceania, and 19.2% in Asia [5]. MDRAB can be resistant to all currently available antibiotics, limiting treatment options and resulting in prolonged hospital stays, excess morbidity and mortality, and a significant economic burden [6]. In this regard, the rise of carbapenem-resistant *Acinetobacter baumannii* (CRAB) has been highly significant, resulting in a globally widespread and alarming problem [7]. In this sense, in 2017, the World Health Organization (WHO) published its first-ever list of antibiotic-resistant “priority pathogens”, among which CRAB was categorized as a critical-priority microorganism [8].

According to the European Antimicrobial Resistance Surveillance Network (EARS-Net), the resistance to carbapenems in invasive isolates during 2021 was 39.9%, and the combined resistance to carbapenems, fluoroquinolones, and aminoglycosides was 36.8% [9]. Other non-exclusively European data sources report 68–82% of CRAB isolates from Saudi Arabia, Egypt, South Africa, Argentina, Brazil, Iran, Pakistan, and Italy [10]; 80–91% of CRAB isolates from Russia, Ukraine, and Belarus [11]; and similarly, 82% CRAB isolates in China [12].

Carbapenem resistance is mainly due to the production of carbapenemases of the OXA-type hydrolyzing class D (class β-lactamases, CHDLs, oxacillinases), the chromosomal carbapenemase OXA-51, and acquisitions such as OXA-23, OXA-24/40, and OXA-58. Less frequently, it is due to class B metallo-β-lactamases such as VIM (Verona Integron-encoded metallo-β-lactamase) and IMP (imipenemase metallo-β-lactamase) [13] or NDM-1 (New Delhi metallo-β-lactamase) [14], the latter of which has recently been noted for the worrying possibility of gene expression without any fitness cost [15]. The co-production of OXA-23 and NDM-1 is infrequent, having been initially detected in African [16,17,18,19,20,21] and Asian [22,23,24,25] countries. In Europe, its presence has only been detected in the Czech Republic and Serbia [26,27].

The main objective of this study was to conduct phenotypic and genotypic characterization of the clonal spread of NDM-1- and OXA-23-co-producing *A. baumannii* isolates for the first time in Spain, which was initiated in a burn unit of a secondary care hospital during a CRAB polyclonal outbreak context.

## 2. Materials and Methods

### 2.1. Study Design

This study was a retrospective cohort observational analytical study.

During the second half of 2022, an increase in the isolation of CRAB isolates with atypical antibiotic susceptibility profiles was noted at the University Hospital Getafe (HUG) in Madrid, Spain. This observation gave us the starting point to initiate the present investigation. HUG is a secondary care hospital with 510 inpatient beds, 18 intensive care beds, 6 burn care beds, and over 1000 hospital admissions per year.

All the patients infected and/or colonized by CRAB isolates between July 2022 and May 2023 were included in this study. The bacterial isolates were obtained from different samples taken from patients admitted to ICU/burn units as well as from different departments in the hospital. An infected case caused by CRAB was defined according to CDC criteria [28], and a colonized case was defined as a patient carrying CRAB without clinical evidence of infection.

A total of 75 samples (41 diagnostic and 34 colonization samples) were collected from the 24 patients affected. One representative CRAB isolate from each patient was selected for phenotypic and genotypic studies, prioritizing isolates implicated in infections according to the criteria established by the CDC, with greater clinical relevance and pathogenic potential.

### 2.2. Identification of Bacterial Isolates and Extraction of Carbapenemases Gene

Presumptive *Acinetobacter* species were isolated on MacConkey agar and sheep blood agar (Becton Dickinson Microbiology Systems, Cockeysville, MD, USA) (https://www.bd.com/es-es/products-and-solutions/products/productfamilies/dehydrated-culture-media-and-additives (accessed on 4 May 2025)).

Bacterial identification was performed using a MALDI-TOF Biotyper instrument (Bruker Daltonics GmbH, Leipzig, Germany) by comparing the unique mass spectra of bacterial proteins with those in a database. All isolates were identified with a score higher than 2, indicating more reliable identification, typically at the species level. Carbapenemase production was tested by a PCR assay (CarbaR+, Novodiag, Hologic, Marlborough, MA, USA), a platform for the multiplex qualitative detection of carbapenemases *bla*_KPC_, *bla*_NDM_, *bla*_VIM_, *bla*_IMP_, *bla*_OXA-23_, *bla*_OXA-24_, *bla*_OXA-48/181_, and *bla*_OXA-58_*,* and of colistin resistance gene *mcr*-1. Both the extraction protocol and PCR conditions were strictly in accordance with the manufacturer’s instructions. The automated software interpretation of the sample results is based on the validation of the internal extraction/inhibition control result (https://www.hologic.com/molecular-diagnostics (accessed on 4 May 2025)). For sample processing, one overnight single colony grown on a MacConkey agar plate was taken with a sterile loop and directly transferred to the eNat preservation medium provided by the manufacturer. The colony was thoroughly suspended, and the eNat tube was vortexed for about 5 s and incubated for 30 min at room temperature for DNA release. After vortexing, 600 µL of the eNat suspension was added to the cartridge that was run on the Novodiag system according to the recommendations (Mobidiag, Espoo, Finland) [29,30]. All the isolates were stored at −80 °C until used.

All the molecular results performed by PCR were confirmed by whole-genome sequencing (WGS) at the national reference laboratory (Centro Nacional de Microbiología, Instituto de Salud Carlos III).

### 2.3. Antibiotic Susceptibility Tests

Antibiotic susceptibility testing (AST) was performed for the selected isolates with a broth microdilution method using the DKMGN Sensititre^TM^ Gram-Negative panels (Thermo Fisher, Waltham, MA, USA), with ATCC 27853 *Pseudomonas aeruginosa* as the quality control strain [31]. In addition, disk diffusion assays were performed for cefiderocol (Oxoid, Thermofisher, Waltham, MA, USA). Susceptibility results for colistin were confirmed by broth microdilution with a UMIC panel (Bruker, Billerica, MA, USA).

All the susceptibility results were interpreted according to European Committee on Antimicrobial Susceptibility Testing (EUCAST) breakpoints [32]. The exceptions were susceptibilities to ceftazidime, cefotaxime, and piperacillin/tazobactam, which were interpreted according to CLSI guidelines [33].

### 2.4. Whole-Genome Sequencing and Read Assembly

Paired-end (2 × 150) libraries were prepared using the Nextera DNA Flex Preparation Kit and sequenced using Illumina HiSeq 500 (Illumina Inc., San Diego, CA, USA) according to the manufacturer’s instructions. Fastp (version 0.23.4) [34] was used to trim the input data that had been assessed using FASTQC (version 0.11.9) for quality (Q20 threshold), followed by de novo assembly using Unicycler (version 0.4.8) [35]. The quality of the assembly was assessed with QUAST (version 5.2.0), and Prokka v1.14 beta was used for automatic genome annotation [36].

### 2.5. Phylogenetic Analysis and Diversity

Ridom SeqSphere+ (version 8.3.1; Ridom, Münsten, Germany) was used to perform core genome Multi-Locus Sequence-Typing analysis (cgMLST), using a built-in scheme for *A. baumannii* containing 2390 core genes, and to construct a minimum spanning tree based on allelic differences. ARIBA (version 2.6.2) [37] was used to determine STs according to the Pasteur (^Pas^) and Oxford (^Oxf^) schemes [38,39].

### 2.6. Antibiotic Resistance Genes, Virulence-Associated Genes, and Plasmids

Antibiotic resistance genes were analyzed via ARIBA (Versión 2-6.2) using the CARD database and ResFinder, with ID thresholds of 100% for β-lactamase variants and 98% for other genes. The presence of acquired resistance genes was considered when the full length of the gene was detected with a mean read depth higher than 25.

The Virulence Finder tool was used to detect genes associated with virulence and their respective sets [40]; PlasmidID was used to map the reads against a curated plasmid database, perform de novo plasmid assemblies, and determine the presence of resistance and replicon genes [41].

The Kaptive (Version 0.0.7–2.0.0) [42] was used to study the capsule polysaccharide K locus (KL) and the outer-core OC locus (OCL virulence-associated genes) of carbapenemase-producing *A. baumannii* isolates.

### 2.7. Insertion Sequences in Carbapenem-Resistant A. baumannii

ISMapper V2.0.2.26 [43] was used to describe copy locations of IS*Aba*1, IS*Aba*10, and IS*Aba*125 [44]. All query sequences were obtained from ISFinder reference sequences. These query sequences, together with paired-end Illumina reads of all isolates and the reference genome (CP010781 https://www.ncbi.nlm.nih.gov/nuccore/CP010781 (accessed on 14 May 2024)), were used as inputs for ISMapper [45].

## 3. Results

### 3.1. Patients and Description of the Outbreak

In July 2022, we detected an unusual increase in the number of cases of infection/colonization with CRAB compared with previous years (Figure 1). The initial analyzed isolates of CRAB came from the Angiology and Vascular Surgery Department, and three weeks later, new CRAB isolates, with similar antibiotic resistance profiles, were detected at the hospital’s burn unit. This detection was related to the admission of four severely burned patients as a consequence of a tanker truck explosion in Libya. Subsequently, we began to detect patients infected/colonized with MDRAB in other hospital wards.

In total, the outbreak affected 24 patients, of which 75 samples were studied in the laboratory, including 41 diagnostic samples (54.6%) and 34 colonization samples (45.4%). Descriptions of the affected patients and temporal outbreak evolution are shown in Table 1 and Figure 1 and Figure 2.

### 3.2. Carbapenemase Types and Phylogenetic Analysis of CRAB Isolates

The carbapenemase genes detected were *bla*_OXA-23_ (21), *bla*_NDM-1_ (14), and *bla*_OXA-72_ (variant of *bla*_OXA-24_) (3); 14 isolates harbored both *bla*_OXA-23_ and *bla*_NDM-1_ (Table 2).

MLST analysis revealed five and six sequence types (STs) according to the Pasteur and Oxford schemes, respectively (Table 2 and Figure 2), with ST1^Pas^/ST231^Oxf^ (14 isolates, 58%) and ST136^Pas^/ST460^Oxf^ (5 isolates, 21%) being the most frequently occurring.

A minimum spanning tree was constructed for all 24 isolates included in this study using the gene-by-gene approach, with allelic distance calculated using cgMLST (Figure 3). Applying a relatedness threshold of five alleles, two groups with more than three related isolates were detected. The average allelic distance between pairs of isolates for both clusters was one allele (Cluster 1 ST1^Pas^/ST231^Oxf^ range: 0–3; Cluster 2 ST136^Pas^/ST460^Oxf^ range: 0–2). The first cluster comprised fourteen isolates producing OXA-23 and NDM-1, and the second group included five isolates producing OXA-23 (Figure 3).

All 14 ST1^Pas^/ST231^Oxf^ OXA-23- and NDM-1-co-producing isolates had the *bla*_OXA-69_ chromosomal carbapenemase gene (a variant of *bla*_OXA-51_), while the ST136^Pas^/ST406^Oxf^ isolates harbored the chromosomal variant *bla*_OXA-409._

Finally, one *bla*_OXA-23-_producing isolate belonged to the ST2^Pas^/ST218^Oxf^. The other two isolates belonged to the ST2^Pas^/ST218^Oxf^ sequence type and were *bla*_OXA-72_ producers indistinguishable by cgMLST, while the other *bla*_OXA-72-_producing isolate belonged to the ST78^Pas^/ST_776_SLV^Oxf^ sequence type.

When analyzing the chronological evolution of the cases (Figure 2), we observed that, before the emergence of the main clone involved in the outbreak (OXA-23/NDM-1; ST1^Pas^/ST231^Oxf^), there were already cases of carbapenemase-producing MDRAB infection belonging to the OXA-72; ST2^Pas^/ST218^Oxf^ clone. This clone, detected for the first time in the Angiology and Vascular Surgery Unit, appeared 6 months later in the same unit, although this new isolate was not an OXA-72 producer but an OXA-23 producer.

During the following month, four patients from Libya were admitted to the burn unit with severe third-degree burns (burns with 40–70% involvement depending on the patient) caused by a tanker explosion. All of them required mechanical ventilation, and CRAB was detected in tracheal aspirates as well as in burn exudates and colonization samples (rectal swab and pharyngeal exudate). Three of them were colonized by CRAB-producing OXA-23 (ST136^Pas^/ST406^Oxf^) at admission to the Burns Unit, and the fourth one was colonized by CRAB, co-producing OXA-23 and NDM-1 (ST1^Pas^/ST231^Oxf^).

### 3.3. Antibiotic Susceptibility Testing of CRAB Isolates

One isolate from each patient involved in the outbreak was tested for AST. All the isolates were resistant to ciprofloxacin, trimethoprim/sulfamethoxazole, aminoglycosides (gentamicin, tobramycin), and carbapenems (imipenem and meropenem). Only one of them was susceptible to amikacin. Likewise, and according to CLSI susceptibility guidelines, all the isolates were resistant to cephalosporins (ceftazidime and cefotaxime) and β-lactam combinations (piperacillin/tazobactam and ampicillin/sulbactam). Overall, 8% of isolates were resistant to colistin (2 out of 24 isolates), and 17% were resistant to cefiderocol (4 out of 24 isolates) (Table 3).

In five patients with an initial colistin-susceptible CRAB isolate, a second colistin-resistant isolate was detected more than fifteen days later. However, after genotypic analysis, no colistin resistance genes (*pmrA* and *pmrB*) were found in these isolates, suggesting that there was a different gene. The same occurred in the case of another patient, in whom the first isolate of CRAB detected was susceptible to cefiderocol and the second isolate detected was resistant, but the iron uptake mutation system (*pir*A/B), which is the most commonly associated mechanism of resistance to this antibiotic, was not detected, nor was the presence of the *bla*_NDM-1_carbapenemases.

All isolates belonging to the ST136^Pas^/ST460^Oxf^ were susceptible to colistin, and only two of them were resistant to cefiderocol. Likewise, the isolates belonging to ST2^Pas^/ST218^Oxf^ (two carriers of *bla*_OXA-72_, the other being *bla*_OXA-23_) were susceptible to both colistin and cefiderocol. Lastly, two of the fourteen isolates belonging to the ST1^Pas^7ST231^Oxf^ sequence type were resistant to colistin, while another two isolates were resistant to cefiderocol. The only amikacin-susceptible (MIC 4 mg/L) isolate belonged to ST315^Pas^/ST231^Oxf^.

### 3.4. Resistome and Virulome of Carbapenemases-Producing CRAB

Resistome analysis included genes associated with acquired resistance to carbapenems, aminoglycosides, sulfonamides, fluoroquinolones, and phenicols, as well as genes associated with chromosomal resistance to β-lactams (cephalosporins and carbapenems). Appendix A shows the acquired antibiotic resistance genes (ARGs) detected, where a mean of 12.2 ARGs was observed (range of 6–15, including acquired carbapenemase genes and excluding chromosomal genes and mutations).

Isolates with dual acquired carbapenemases (*bla*_OXA-23_/*bla*_NDM-1_ (*n* = 14)) had more ARGs (mean = 14.6; range 14–15) than isolates with one acquired carbapenemase (*bla*_OXA-23_; *n* = 9 and *bla*_OXA-72_; *n* = 1) (mean = 8; range: 6–10).

Chromosomal constitutive genes, *bla*_OXA-69_ carbapenemases and *bla*_ADC-191_ betalactamases, were detected in ST1^Pas^/ST231^Oxf^ CRAB isolates, while *bla*_OXA-409_ carbapenemases and *bla*_ADC-88_ betalactamases were detected in ST136^Pas^/ST460^Oxf^ isolates.

No acquired resistance genes, encoding resistance to fluoroquinolones, were detected. Isolates belonging to ST136^Pas^/ST406^Oxf^ expressed the mutations *gyrA* codon 81 TCA (Ser) → TTA (Leu) and *parC* codon 84 TCA (Ser) → TTA (Leu), whereas in ST1^Pas^/ST231^Oxf^ isolates only *gyrA* codon 81 TCA (Ser) → TTA (Leu) mutations were found (Appendix A).

Regarding non β-lactamase ARGs, 100% of the genes analyzed were detected with aminoglycoside resistance, the most predominant being *aac (3)-IIa*, (100%; *n* = 24); *ant (2``)-Ia* (83.3%, *n* = 20) and *ant (3″)-Ia* (50%, *n* = 12). No association was found between the aminoglycoside resistance genes and the sequence types of the isolates. In addition, the 16s rRNA methylase gene *armA,* which confers resistance to all aminoglycosides, was present in 71% (*n* = 17) of the isolates, including all the isolates belonging to the ST1^Pas^/ST231^Oxf^ cluster (Appendix A).

Resistance to trimethoprim-sulfamethoxazole was associated with the *sul1*, *sul2*, and *dfrA1* genes. All ST1^Pas^/ST231^Oxf^ isolates produced *sul1/sul2* and *sfrA1* genes, while only the *sul2* gene was detected in ST136^Pas^/ST406^Oxf^ and ST2^Pas^/ST218^Oxf^ isolates (Appendix A).

The analysis also includes information on the role of upstream insertion sequences disrupting the outer membrane protein gene *carO* as an additional mechanism of resistance to carbapenems due to their nonspecific and passive diffusion properties [46]. The CRAB isolates, affected by *car*O disruption resulting from insertion sequences IS*Aba125* and IS*Aba10,* are shown (Appendix A). All the isolates belonging to ST1^Pas^/ST231^Oxf^ showed disruptions in *car*O mediated by IS*Aba*10, and this was also true for 10 out of 14 isolated mediated by IS*Aba*125. No IS*Aba1*-mediated disruption was observed in any of the isolates analyzed.

### 3.5. Characterization of the Virulence-Associated Genes

In the 24 genomes of *A. baumannii*, 17 genes associated with virulence factors were studied (Appendix A). All isolates presented the 17 virulence genes, including the gene encoding *Omp*A, which is involved in host cell adhesion and invasion [47]; the *aba1* inducer, which is involved in quorum sensing [48]; and the *pga*ABC locus, which is associated with polysaccharide biosynthesis and biofilm formation [49]. Virulence genes responsible for iron uptake through the production of the siderophore acinetobactin *ent*E [50] and the *csu*A/B ABCDE operons were involved in pili synthesis and assembly [51].

### 3.6. Capsular Exopolysaccharide in CRAB Isolates

We also determined the capsular polysaccharide K locus (KL) and the outer-core OC locus (OCL) types. Six types of K locus were detected, where the main types were KL17 (*n* = 14, 58%) and KL25 (*n* = 5, 21%). All isolates with KL7 belonged to the ST1^Pas^/ST231^Oxf^ sequence type. The KL25 type was expressed by isolates belonging to the ST136^Pas^/ST406^Oxf^ sequence type.

The other four K locus types were KL9 (in ST2^Pas^/ST218^Oxf^ isolates producers of OXA-72 carbapenemases), KL7 (in ST2^Pas^/ST218^Oxf^ isolate producer of OXA-23 carbapenemases), KL91 (in ST315^Pas^/ST231^Oxf^ isolate), and KL3 (in ST78^Pas^/ST944^Oxf^ isolate). As for the outer-core capsule, OCL1 was the most common type of capsule (15 isolates, 63%). Fourteen out of fifteen isolates belonged to the ST1^Pas^/ST231^Oxf^ sequence type, while one isolate belonged to the ST78^Pas^/ST944^Oxf^ sequence type. OCL3 was detected in all the ST136^Pas^/ST406^Oxf^ sequence-type isolates. The remaining isolates exhibited variable OCLs (Appendix A). Both KLs and OCLs exhibited a coverage of 100% and an identity above 98%.

### 3.7. Detection of Plasmids in CRAB Isolates

Plasmids were detected in 21 of 24 CRAB isolates. As seen in the results, the pS32-1 plasmid was detected in 13 of 14 isolates belonging to the ST1^Pas^/ST231^Oxf^ clone. Another plasmid was only detected in one of them (pA297-1 (pRAY*)). The most common plasmid replicon was R3-T1, detected in 14 isolates (13 of which belonged to clone ST1^Pas^/ST231^Oxf^). pD4 and pD72-2 plasmids were detected in ST136^Pas^/ST406^Oxf^ isolates (Appendix A).

### 3.8. Infection Control Measures and Outcome

Infection prevention and control strategies aimed at preventing the spread of these microorganisms included contact precautions, daily chlorhexidine baths, patient cohorting, environmental disinfection, and active rectal screening. In addition, we decided to look for a reservoir that could explain the rapid dissemination of microorganisms. For this purpose, surface samples were taken at critical points in the different ICU and burn boxes, but no conclusion could be reached. Finally, an effective and aggressive intervention during the last quarter of 2022, in which the patients were cohorted, was necessary, and an exhaustive cleaning of the units involved was carried out. These measures were sufficient to resolve the outbreak; no more cases of CRAB, co-producing OXA-23 and NDM-1, (ST1^Pas^/ST231^Oxf^) were detected, and the general incidence fell to pre-outbreak levels (Figure 1).

## 4. Discussion

In this study, we describe the clonal dissemination of the OXA-23- and NDM-1-co-producing CRAB ST1 clone in a Spanish hospital after the admission of a severely burned patient from Libya infected with this bacterium. This co-production of carbapenemases is an uncommon combination that could generate diagnostic and therapeutic challenges. There are some descriptions of NDM-1- and OXA-23-co-producing CRAB isolates in African, Asian, and European countries, but they mainly implicate sporadic isolates that have been emerging recently [16,17,18,19,20,21,22,23,24,25,26,27]. To the best of our knowledge, this is the first description of this CRAB genotype in Spain following cross-border dissemination.

The incidence of infections caused by MDRAB in HUG has been infrequent during the last 10 years, with fewer than 10 cases per year of infected/colonized patients detected (for an average of 7 cases/year). A unique significant outbreak was detected during 2016–2017, which involved 43 ICU patients (Figure 1) and was largely caused by an OXA-23/ST2^Pas^-ST2164^Oxf^ cluster, unrelated to the current one; the outbreak was confined to the ICU, and no further cases were detected in the rest of the hospital.

The event described in the present study, which occurred in the second half of 2022, showed a 20-fold increase in incidence compared to the previous 4-year period, with a prevalence of MDRAB strains close to 75%. Unlike what occurred in 2016–2017, it affected multiple units. This inter-unit dissemination could have occurred as a consequence of the need for these patients to heal their burns, involving a transfer to different units. The dissemination capacity of *A. baumannii* among medical facilities, mainly due to its ability to persist on dry surfaces and to acquire resistance to different classes of antibiotics, is well documented [52].

It is important to highlight the importance of integrating whole-genomic sequencing into CRAB surveillance, as advised by the ECDC, and for which coordination with national reference laboratories takes on special importance [53].

Numerous studies have recently reports outbreaks caused by MDRAB with carbapenemase production in ICU and major burn units [54,55,56,57,58,59]. MDRAB is among the ten most frequently isolated microorganisms in ICU-acquired healthcare-associated infections [55]. A recent study provided a global view on CRAB, showing that the situation in Europe reflects an increase in these kinds of strains, among which the production of the metallo-beta-lactamases, although rare, is gaining some importance [60].

Regarding AST, the results indicate that NDM-1 is detected in both colistin-resistant and colistin-susceptible isolates belonging to ST1^Pas^/ST231^Oxf^, and in ST2^Pas^/ST218^Oxf^, it is detected in colistin-susceptible isolates. This finding is not in agreement with recent studies in which NDM is only detected in colistin-resistant ST1 isolates [61]. Cefiderocol is an alternative in treatment for MDRAB infections. Guidance documents from various American and European scientific societies recommend cefiderocol for treating CRAB infections. Our results show that 17% of the isolates analyzed were resistant to this antibiotic. These results are in agreement with previous studies describing the decreased efficacy of this antibiotic in MDRAB isolates [62].

We also detected Omp*A* protein in 100% of the isolates. It is known that this protein plays various roles related to virulence and bacteria’s survival under harsh conditions, such as adhesion, invasion, apoptosis, and antibiotic resistance. It also plays an important role in biofilm formation [63]. All isolates, except one, were resistant to the aminoglycosides gentamicin, tobramycin, and amikacin. In the amikacin-susceptible isolate (MIC ≤4 mg/L), no genes such as *aph (3)-VI* or 16s rRNA *armA* methylase gene, both associated with amikacin resistance, were detected [64,65]. The *armA* methylase gene was detected in all ST1^Pas^/ST231^Oxf^ isolates. These findings are consistent with previous reports describing the co-occurrence of 16S rRNA methylase ArmA with *bla_NDM_*_-1_ and *bla_OXA_*_-23_ in *A. baumannii* clinical isolates [66].

Some results related to resistance to aminoglycosides in strains co-producing OXA-23 and NDM carbapenemases have been recently reported. This association has been observed in MDRAB clinical isolates from Egypt, although our results differ from those reported by these investigators since they describe the greater variability of high-risk clones [67]. Although not all ST1 isolates showed the same mechanism of resistance to aminoglycosides, the same mechanism of resistance to the rest of the antibiotics studied (excluding beta-lactams) was detected in all the ST1 isolates, with quinolone resistance being related to the expression of the mutation in the *gyrA*_S81L gene, trimethoprim-sulfamethoxazole resistance being related to the presence of *sul1*/*sul2* and *dfrA1* genes, and *cmlA5* gene expression (chloramphenicol resistance) and *sat2* gene expression being related to resistance to macrolides, lincosamines, and streptogramins. Regarding colistin resistance, no resistance genes to this antibiotic were detected in isolates that showed a change in colistin susceptibility. This could be due to adaptive resistance, probably because of the use of colistin to treat these patients [68].

In this case, the infection control measures carried out were sufficient to manage and control the outbreak. These measures were previously proven to be the most effective measures for permanently eliminating the spread of MDRAB [69], although we must be aware that the problem may increase in the coming years.

The cross-border dissemination of MDRAB high-risk clones may increase over time as we are witnessing a worldwide increase in the incidence of MDRAB infection, with Asia and Africa being the most affected continents [70,71,72,73]. This is likely to lead to a higher probability of spread to other countries such as those in Europe [74,75,76,77]. In this case, the index case was a patient from Libya, an African country of the Arab League, where there is a high prevalence of MDRAB infections, reaching 88% of the multiresistant isolates studied [70]. It is not only important to know the prevalence of these microorganisms in other countries, but also to take into account that there are certain events that increase frequency, such as wars that impose direct consequences such as the movement of refugees or evacuated patients from the country involved, which can lead to a change in the local ecology with the emergence of previously undetected multidrug-resistant microorganisms [78]. European structured surveys, including WGS analysis, that allow for the identification of successful clones of CRAB and the extent of their spread provide a better understanding of predominant resistance mechanisms to carbapenems and detect potential cross-border spread [53].

In conclusion, this study documents the clonal spread of an emerging NDM-1- and OXA-23-co-producing *A. baumannii* strain in Spain, belonging to the high-risk clone ST1 and linked to a Libyan patient. This highlights the risk of cross-border spread of multidrug-resistant microorganisms. Although infection control measures were successful in containing the outbreak, the integration of whole-genome sequencing into CRAB surveillance in coordination with national reference laboratories was essential.

It is important to maintain surveillance strategies as the incidence of CRAB infections is expected to increase in the coming years.

## Figures and Tables

**Figure 1 microorganisms-13-01149-f001:**
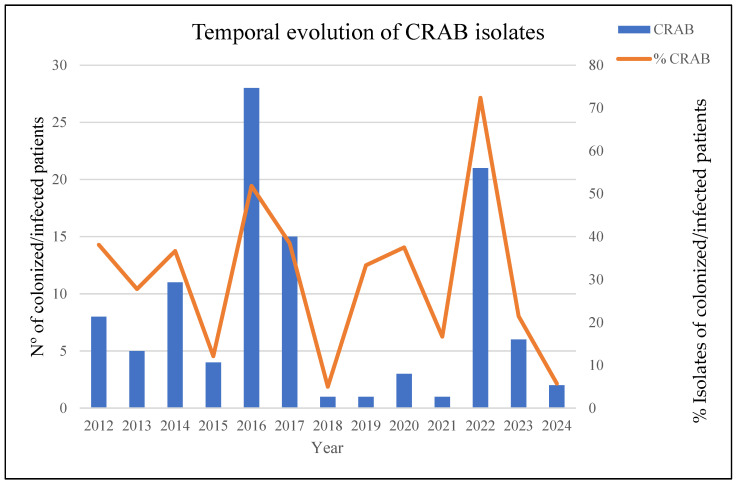
The temporal evolution of CRAB-colonized/infected patients over the last 12 years: left axis—absolute number of CRAB isolates; right axis—percentage of CRAB isolates relative to total *A. baumannii* isolates.

**Figure 2 microorganisms-13-01149-f002:**
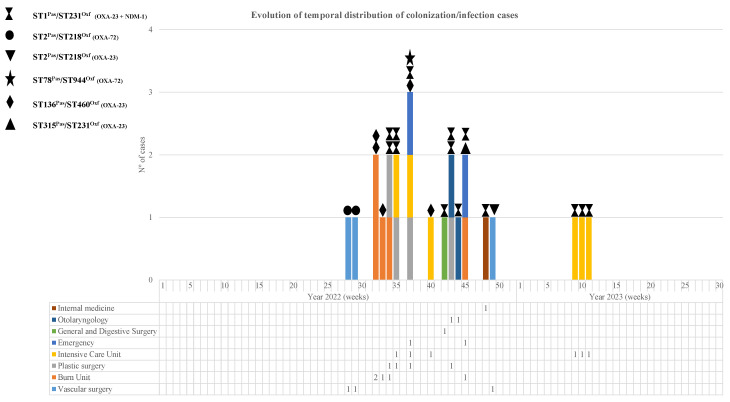
Weekly temporal distribution of CRAB sequencing types (STPas/STOxf) and carbapenemase types during the study period.

**Figure 3 microorganisms-13-01149-f003:**
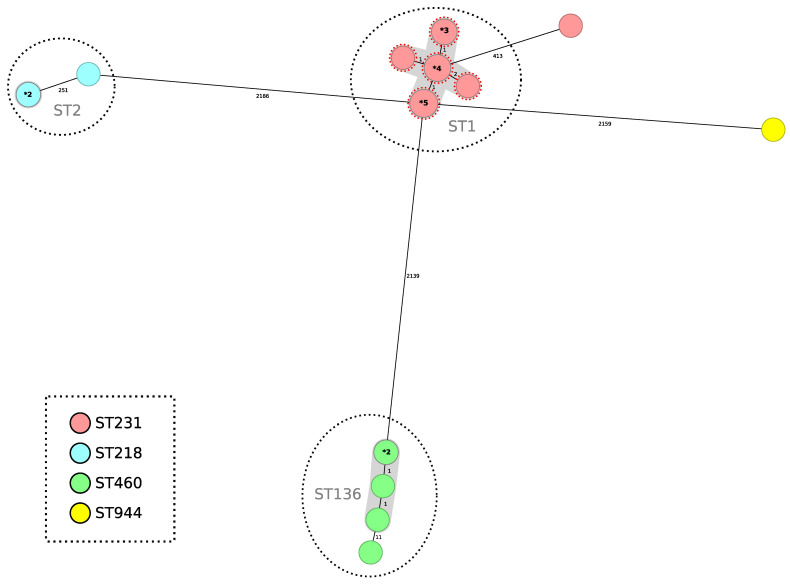
Minimum spanning tree representing distances between *A. baumannii* genomes by applying cgMLST of 2390 genes. Colors in each circle indicate ST^Oxf^ ST and gray ovals represent groups (more than two isolates) with the same ST^Pas^. A dotted red circle indicates that the strain has NDM-1. Where a circle corresponds to more than one isolate, the number of isolates is indicated in bold type preceded by an asterisk. Gray shadows represent a cluster of strains; a threshold of 5 alleles was applied.

**Table 1 microorganisms-13-01149-t001:** Chronological order of representative CRAB isolates selected from each patient.

Microbiological Data of Patients						
Case	Age	Sex (M/F) *	Date of First Isolation	Sample	Diagnostic/Colonization	Clinical Ward Where the Sample Was Obtained
1	72	M	7 July 2022	Biopsy	Diagnostic	Vascular surgery
2	89	M	14 July 2022	Abscess/Pus	Diagnostic	Vascular surgery
3	35	M	6 August 2022	Burn exudate	Colonization	Burn unit
4	24	M	6 August 2022	Tracheal aspirate	Diagnostic	Burn unit
5	19	M	14 August 2022	Tracheal aspirate	Diagnostic	Burn unit
6	32	M	15 August 2022	Tracheal aspirate	Diagnostic	Burn unit
7	50	M	17 August 2022	Catheter	Diagnostic	Plastic surgery
8	50	M	25 August 2022	Catheter	Diagnostic	Plastic surgery
9	78	M	25 August 2022	Blood	Diagnostic	Intensive care unit
10	68	M	5 September 2022	Burn exudate	Diagnostic	Plastic surgery
11	84	F	9 September 2022	Aspirate puncture	Diagnostic	Intensive care unit
12	82	F	11 September 2022	Surgical wound exudate	Diagnostic	Emergency
13	90	M	28 September 2022	Blood	Diagnostic	Intensive care unit
14	46	F	13 October 2022	Rectal swab	Colonization	General and digestive surgery
15	78	M	19 October 2022	Abscess/Pus	Diagnostic	Otolaryngology
16	79	F	21 October 2022	Rectal swab	Colonization	Plastic surgery
17	63	F	27 October 2022	Rectal swab	Colonization	Otolaryngology
18	61	F	3 November 2022	Rectal swab	Colonization	Burn unit
19	43	F	6 November 2022	Skin and soft tissue exudate	Diagnostic	Emergency
20	19	F	24 November 2022	Rectal swab	Colonization	Internal medicine
21	57	F	1 December 2022	Non-surgical wound exudate	Diagnostic	Vascular surgery
22	52	F	21 February 2023	Rectal swab	Colonization	Intensive care unit
23	78	M	28 February 2023	Tracheal aspirate	Diagnostic	Intensive care unit
24	64	F	3 March 2023	Tracheal aspirate	Diagnostic	Intensive care unit

* M: male. F: female.

**Table 2 microorganisms-13-01149-t002:** Clonal lineages (Pasteur and Oxford schemes) and β-lactamase genes were identified through sequencing experiments.

Isolate	Pasteur ST	Oxford ST	Acquired β-Lactamase	Chromosomal β-Lactamase	
			Carbapenemase	Carbapenemase	AmpC
1	2	218	*bla* _OXA-72_	*bla* _OXA-66_	*bla* _ADC-30_
2	2	218	*bla* _OXA-72_	*bla* _OXA-66_	*bla* _ADC-30_
3	136	460	*bla* _OXA-23_	*bla* _OXA-409_	*bla* _ADC-88_
4	136	460	*bla* _OXA-23_	*bla* _OXA-409_	*bla* _ADC-88_
5	136	460	*bla* _OXA-23_	*bla* _OXA-409_	*bla* _ADC-88_
6	1	231	*bla* _OXA-23_ *bla* _NDM-1_	*bla* _OXA-69_	*bla* _ADC-191_
7	1	231	*bla* _OXA-23_ *bla* _NDM-1_	*bla* _OXA-69_	*bla* _ADC-191_
8	1	231	*bla* _OXA-23_ *bla* _NDM-1_	*bla* _OXA-69_	*bla* _ADC-191_
9	1	231	*bla* _OXA-23_ *bla* _NDM-1_	*bla* _OXA-69_	*bla* _ADC-191_
10	136	460	*bla* _OXA-23_	*bla* _OXA-409_	*bla* _ADC-88_
11	1	231	*bla* _OXA-23_ *bla* _NDM-1_	*bla* _OXA-69_	*bla* _ADC-191_
12	78	944	*bla* _OXA-72_	*bla* _OXA-90_	*bla* _ADC-152_
13	136	460	*bla* _OXA-23_	*bla* _OXA-409_	*bla* _ADC-88_
14	1	231	*bla* _OXA-23_ *bla* _NDM-1_	*bla* _OXA-69_	*bla* _ADC-191_
15	1	231	*bla* _OXA-23_ *bla* _NDM-1_	*bla* _OXA-69_	*bla* _ADC-191_
16	1	231	*bla* _OXA-23_ *bla* _NDM-1_	*bla* _OXA-69_	*bla* _ADC-191_
17	1	231	*bla* _OXA-23_ *bla* _NDM-1_	*bla* _OXA-69_	*bla* _ADC-191_
18	315	231	*bla* _OXA-23_	*bla* _OXA-69_	*bla* _ADC-79_
19	1	231	*bla* _OXA-23_ *bla* _NDM-1_	*bla* _OXA-69_	*bla* _ADC-191_
20	1	231	*bla* _OXA-23_ *bla* _NDM-1_	*bla* _OXA-69_	*bla* _ADC-191_
21	2	218	*bla* _OXA-23_	*bla* _OXA-69_	*bla* _ADC-30_
22	1	231	*bla* _OXA-23_ *bla* _NDM-1_	*bla* _OXA-69_	*bla* _ADC-191_
23	1	231	*bla* _OXA-23_ *bla* _NDM-1_	*bla* _OXA-69_	*bla* _ADC-191_
24	1	231	*bla* _OXA-23_ *bla* _NDM-1_	*bla* _OXA-69_	*bla* _ADC-191_

**Table 3 microorganisms-13-01149-t003:** Antibiotic susceptibility of 24 carbapenemase-producing Acinetobacter baumannii isolates as determined by the microdilution method according to the European Committee on Antimicrobial Susceptibility Testing (EUCAST) and Clinical and Laboratory Standards Institute (CLSI).

Antibiotics.	* S% Total Isolates (*n*)	* I% Total Isolates (*n*)	* R% Total Isolates (*n*)	*** MIC_50_	*** MIC_90_
Amikacin	4.2 (1)	-	95.8 (23)	32	32
Cefiderocol **	83.3 (20)	-	16.7 (4)	-	-
Ceftazidime	0	-	100 (24)	>16	>16
Ciprofloxacin	0	-	100 (24)	>2	>2
Colistin	91.6 (22)	-	8.3 (2)	1	1
Gentamicin	0	-	100 (24)	>8	>8
Imipenem	0	-	100 (24)	>16	>16
Meropenem	0	-	100 (24)	>16	>16
Piperaciclin/Tazobactam	0	-	100 (24)	>32/4	>32/4
Tobramycin	0	-	100 (24)	>8	>8
Trimethoprim-Sulfamethoxazole	0	12.5 (3)	87.5 (21)	>8/152	>8/152

* S: susceptibility. * R: resistance. * I: susceptible increased exposure. ** disk diffusion method (30 µg) on plate. It is considered susceptible with a zone diameter of ≥17 mm according to PK-PD breakpoint. *** MIC50: minimum inhibitory concentration (MIC) to inhibit the growth of 50% of the tested microorganisms. MIC90: minimum inhibitory concentration (MIC) to inhibit the growth of 90% of the tested microorganisms.

## Data Availability

The original contributions presented in this study are included in the article/Appendix A. Further inquiries can be directed to the corresponding authors.

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
