# Peer review of "Emergence of NDM-1- and OXA-23-Co-Producing Acinetobacter baumannii ST1 Isolates from a Burn Unit in Spain"

_microorganisms, 2025, doi:10.3390/microorganisms13051149_

Round 1

Reviewer 1 Report

Comments and Suggestions for Authors

The manuscript “Emergence of NDM-1 and OXA-23 Co-Producing Acinetobacter baumannii ST1 isolates from a Burn Unit in Spain” by Hidalgo et al. describes the emergence in Spain of a carbapenem-resistant Acinetobacter baumannii (CRAB) strain that coproduces the enzymes NDM-1 and OXA-23. Analysis of the strains revealed resistance to multiple antibiotics and the presence of diverse types of genetic sequences, suggesting clonal dissemination. The infection control measures implemented successfully contained the outbreak, but the study highlights the risk of cross-border spread of these resistant bacteria, possibly linked to patients from regions with a high prevalence of these strains.

While the article presented holds promise, aspects of it would benefit from additional refinement to enhance their robustness and rigor. Specific areas for improvement are outlined below:

Abstract

Line 18: “In the summer of 2002…”. Is it 2002 or 2022? Please correct the text.

Introduction:

Lines 43-44: “The rapid emergence of multidrug-resistant Acinetobacter baumannii (MDRAB) has led to a worrying situation worldwide”. Before this sentence the authors could mention some aspects about the biology of Acinetobacter baumannii. In this section, it would be advisable for the authors to explain the importance of studying multidrug-resistant strains in the context of public health.

Lines 47-48: “A multinational study of ICUs revealed that the prevalence of MDRAB was 14.8% in Africa….”. Please, expand on the consequences of CRAB infections. Including increased mortality, prolonged hospital stays, higher treatment costs, and limitations in therapeutic options.

Lines 69-70: “In Europe, its presence has only been detected in two countries”. indicate in the text which are those two countries. Integrate this into the previous paragraph of text.

Lines 69-70: “In Europe, its presence has only been detected in two countries”. indicate in the text which are those two countries. Integrate this into the previous paragraph of text.

Materials and Methods:

Line 72: “Study Design”. Please indicate the type of study design.

Lines 89-90: “One representative CRAB isolates from each patient was selected for phenotypic and genotypic studies, prioritizing isolates implicated in infections”. It is recommended to indicate all the criteria used to prioritize the isolates in cases where multiple isolates were available.

Lines 93-94: “Initial identification was performed by MALDI-TOF”. The authors could briefly describe the procedure and criteria used.

Lines 94-95: “Carbapenemase production was tested by a PCR assay”. I recommend that the authors describe how DNA extraction and quantification were performed, as well as the conditions used in PCR preparation and amplification. A table indicating the sequence of the primers used could also be included as supplementary material.

Line 105: Why was Pseudomonas aeruginosa used as a control when the study focused on Acinetobacter baumannii strains?

Lines 116-117: “The Quality of the reads was assessed using FASTQC”. Was the reads not trimming before assembly?

Line 117: “de novo”. Please, put in italics.

Line 119: “de novo assembly annotation”. Replace with genome annotation.

Lines 127-129: “Antibiotic resistance genes were analyzed by ARIBA (Versión 2-6.2) using the CARD database and ResFinder with ID thresholds of 100% for β-lactamase variants and 98% for other genes”. Was a minimum coverage threshold considered?

Lines 130-131: “The Virulence Finder tool was used to detect genes associated with virulence and their respective sets”. Was some type of identity and coverage threshold used in this analysis?

Comments on the Quality of English Language

The manuscript requires revision by a native language specialist to address grammatical errors and improve paragraph structure.

Author Response

Response to Reviewer 1 Comments

Point-by-point response to Comments and Suggestions for Authors

Comments 1:  Abstract Line 18: “In the summer of 2002…”. Is it 2002 or 2022? Please correct the text.

Response 1: We agree with this comment. It is a mistake.

Corrected in the manuscript.

Comments 2: Introduction. Lines 43-44: “The rapid emergence of multidrug-resistant Acinetobacter baumannii (MDRAB) has led to a worrying situation worldwide”. Before this sentence the authors could mention some aspects about the biology of Acinetobacter baumannii. In this section, it would be advisable for the authors to explain the importance of studying multidrug-resistant strains in the context of public health.

Response 2: We agree with this comment. We have, accordingly, modified the beginning of the Introduction section by adding a paragraph to emphasize this point. We have also added new references. The new paragraph reads as follows:

“Antimicrobial resistance is emerging as one of the most important threats to global public health, threatening the effective treatment of an increasing number of bacterial infections in various healthcare settings (Restrepo-Arbeláez, 2024). Multidrug-resistant Acinetobacterbaumannii (MDRAB) has been identified as one of the most important pathogens with high rates of resistance to multiple classes of antibiotics (Morris S Antibiotics 2020). Recent genomic and phenotypic analyses of A. baumannii have identified several virulence factors responsible for its pathogenicity, including porins, capsular polysaccharides, lipopolysaccharides, phospholipases, outer membrane vesicles, metal acquisition systems, and protein secretion systems (Chang-Ro Lee 2017 Frontiers in cellular and Infection microbiology, McConnell et al, 2013 FEMS Microbiol. Rev).”

Comments 3. Introduction. Lines 47-48: “A multinational study of ICUs revealed that the prevalence of MDRAB was 14.8% in Africa….”. Please, expand on the consequences of CRAB infections. Including increased mortality, prolonged hospital stays, higher treatment costs, and limitations in therapeutic options.

Response 3: We agree with this comment. We have modified the paragraph accordingly and also added new references as follows:

“MDRAB can be resistant to all currently available antibiotics, limiting treatment options and resulting in prolonged hospital stays, excess morbidity and mortality, and a significant economic burden (C.D.C. Us. Antibiotic resistance threats in the United States, 2019. In: Centers for Disease C, Prevention, National Center for Emerging Z et al.; 2019. Atlanta, GA.)

Comments 4. Introduction Lines 69-70: “In Europe, its presence has only been detected in two countries”. indicate in the text which are those two countries. Integrate this into the previous paragraph of text.

Response 4:  We agree with this comment. We have modified the paragraph accordingly:

“In Europe, its presence has only been detected in Czech Republic and Serbia”

Comments 5. Materials and Methods Line 72: “Study Design”. Please indicate the type of study design.

Response 5: We agree with this comment. We have included the paragraph accordingly:

“This study was a retrospective cohort observational analytical study.”

Comments 6. Materials and Methods Lines 89-90: “One representative CRAB isolates from each patient was selected for phenotypic and genotypic studies, prioritizing isolates implicated in infections”. It is recommended to indicate all the criteria used to prioritize the isolates in cases where multiple isolates were available.

Response 6: Table 1 shows which isolates were selected in this study. In the Materials and Methods section, it is explained that clinical isolates are prioritized over colonization isolates according to the criteria established by the CDC, because of their greater clinical relevance and pathogenic potential. The isolates listed as colonization isolates in Table 1 indicate that no clinical CRAB isolates were obtained from that patient.

We have modified the paragraph and title of Table 1 for clarity.

One representative CRAB isolate from each patient was selected for phenotypic and genotypic studies, prioritizing isolates implicated in infections according to the criteria established by the CDC, greater clinical relevance and pathogenic potential”

“Table 1: Chronological order of representative CRAB isolates selected from each patient.”

Comments 7. Materials and Methods Lines 93-94: “Initial identification was performed by MALDI-TOF”. The authors could briefly describe the procedure and criteria used.

Response 7: We agree with this comment. We have modified the paragraph for clarity.

“Bacterial identification was performed using a MALDI-TOF Biotyper instrument (Bruker Daltonics GmbH, Leipzig, Germany) by comparing the unique mass spectra of bacterial proteins with those in a database. All isolates were identified with a score higher than 2, indicating a more reliable identification, typically at the species level.”

Comments 8. Materials and Methods Lines 94-95: “Carbapenemase production was tested by a PCR assay”. I recommend that the authors describe how DNA extraction and quantification were performed, as well as the conditions used in PCR preparation and amplification. A table indicating the sequence of the primers used could also be included as supplementary material.

Response 8: The PCR assay was not developed in-house; accordingly, both the extraction protocol and PCR conditions were strictly followed in accordance with the manufacturer’s instructions.

For clarity we have included the following paragraph and added new references  as follows :

“Carbapenemase production was tested by a PCR assay (CarbaR+, Novodiag, Hologic), a platform for the multiplex qualitative detection of carbapenemases blaKPC, blaNDM, blaVIM,blaIMP, blaOXA-23, blaOXA-24, blaOXA-48/181, blaOXA-58 and colistin resistance gene mcr-1.Both, the extraction protocol and PCR conditions were strictly followed in accordance with the manufacturer’s instructions. The automated software interpretation of the sample results is based on the validation of internal extraction/inhibition control result. For sample processing, one overnight single colony grown on MacConkey agar plate was taken with a sterile loop and directly transferred in eNat preservation medium provided by the manufacturer. The colony was thoroughly suspended and the eNat tube vortexed for about 5 seconds and incubated for 30 minutes at room temperature for DNA release. After vortexing, 600 µL of the eNat suspension were added to the cartridge that was run on the Novodiag system according to the manufaturer´srecommendations (Mobidiag, Espoo, Finland) (Microb Drug Resist. 2021 Feb;27 (2):170-178) (J Microbiol Methods. 2021 Jan:180:105-106).”

Comments 9. Materials and Methods Line 105: Why was Pseudomonas aeruginosa used as a control when the study focused on Acinetobacter baumannii strains?

Response 9: Pseudomonas aeruginosa strain ATCC 27853 can be used for quality control in Acinetobacter baumannii susceptibiliity testing.  

ATCC 27853, a strain of Pseudomonas aeruginosa, is commonly used as a quality control (QC) strain in various applications, particularly in antimicrobial susceptibility testing. It's a well-characterized strain, facilitating the reproducibility and accuracy of testing results across different laboratories. 

It is used in the quality control of products from different manufacturers, including Abbott, API, Autobac, BBL, bioMerieux Vitek, Micro-Media, MicroScan, Roche Diagnostics, and Sensititre. 

Pseudomonas aeruginosa strain ATCC 27853 can be used for quality control in Acinetobacter baumannii susceptibiliity testing. This strain is a recognized standard for this purpose, and is commonly used in conjunction with Escherichia coli strain ATCC 25922.

Enclosed two publications

Antimicrob Agents Chemother. 2009 Sep;53(9):3628-34.doi: 10.1128/AAC.00284-09. Epub 2009 Jun 15.

J Infect Public Health. 2018 Nov-Dec;11(6):856-860. doi: 10.1016/j.jiph.2018.07.006. Epub 2018 Jul 26.

Comments 10. Materials and Methods Lines 116-117: “The Quality of the reads was assessed using FASTQC”. Was the reads not trimming before assembly?

Response 10: Reads have been trimmed with Fastp software, and this has been modified in the paragraph and added a new reference as follows:

“Fastp (version 0.23.4) (Chen S, Zhou Y, Chen Y, Gu J. fastp: an ultra-fast all-in-one FASTQ preprocessor. Bioinformatics. 2018 Sep 1;34(17): i884-i890. doi: 10.1093/bioinformatics/bty560. PMID: 30423086; PMCID: PMC6129281) has been used to trim input data for quality (Q20 threshold) that has been assessed using FASTQC (version 0.11.9), followed by de novo assembly using Unicycler (version 0.4.8)[29]. The quality of the assembly was assessed with QUAST (version 5.2.0), and Prokka v1.14 beta was used for automatic de novo assembly annotation [30]”

.

Comments 11. Materials and Methods Line 117: “de novo”. Please, put in italics.

Response 11: We agree with this comment. Corrected in the manuscript.

Comments 12. Materials and Methods Line 119: “de novo assembly annotation”. Replace with genome annotation.

Response 12: We agree with this comment. Corrected in the manuscript.

Comments 13: Material and Methods Lines 127-129: “Antibiotic resistance genes were analyzed by ARIBA (Versión 2-6.2) using the CARD database and ResFinder with ID thresholds of 100% for β-lactamase variants and 98% for other genes”. Was a minimum coverage threshold considered?

Response 13: We agree with this comment. We have added the paragraph as follows:

“Antibiotic resistance genes were analyzed by ARIBA (Versión 2-6.2) using the CARD database and ResFinder with ID thresholds of 100% for β-lactamase variants and 98% for other genes. Presence of acquired resistance genes was considered when the full length of gene was detected with a mean read depth higher than 25.”

Comments 14: Material and Methods Lines 130-131: “The Virulence Finder tool was used to detect genes associated with virulence and their respective sets”. Was some type of identity and coverage threshold used in this analysis?

Response 14: The Virulence Finder tool was used to detect genes associated with virulence and their respective sets [34]. ID threshold and minimum length selected for Virulence Finder were 90% and 60% respectively.

Response to Comments on the Quality of English Language

Point 1: The manuscript requires revision by a native language specialist to address grammatical errors and improve paragraph structure.

Response 1: We will send the manuscript for language review by a native specialist.

Reviewer 2 Report

Comments and Suggestions for Authors

Dear authors, my comments are presented below:

  1. References [8] and [9] - lines 143-144 – obscure sources. Please specify the sources in accordance with the rules of the bibliographic list design.
  2. All abbreviations used in tables and figures must be deciphered, for example, Table 2 - MIC50 – there is no transcription of the abbreviation in the signature.
  3. Line 21 – “...NDM-1 and OXA-23...” – requires a transcription of the abbreviation at the first mention in the text.
  4. Line 18 - “In the summer of 2002…” – you write about the outbreak of the disease in 2002, but then you say that (line 23) “Patients with multidrug-resistant baumannii isolates (July 2022-May 2023)…” – only 20 years after the outbreak of the disease, have they started studying pathogenic multidrug-resistant A. baumannii isolates? And in the Discussion section, the authors describe an outbreak that occurred in 2016-2017. Please specify which event is in question in which year? And in what years were the samples taken?
  5. Line 92 - “Presumptive Acinetobacter species were isolated on MacConkey agar…” – you need to add either a link or the composition of the medium.
  6. Line 91 – Section “2.2. Identification of Bacterial Isolates and Extraction of Carbapenemases Gene” – In the section, you should either add links to the listed methods with a brief description, or describe the techniques in detail.
  7. It is not clear why the tables and figures are located at the end of the Results section, and not after the mention in the text during the presentation of the results.
  8. How many isolates were allocated in total for further work.
  9. There are no statistical data in the work.
  10. Since the authors have the results of genome-wide sequencing, it would not be superfluous to add a genome-wide phylogenetic tree of the studied isolates.
  11. Please add the Conclusion section.

Author Response

Response to Reviewer 2 Comments

Point-by-point response to Comments and Suggestions for Authors

Comments 1: References [8] and [9] - lines 143-144 – obscure sources. Please specify the sources in accordance with the rules of the bibliographic list design.

Response 1: Thank you for pointing this out. We have, accordingly, modified

Comments 2: All abbreviations used in tables and figures must be deciphered, for example, Table 2 - MIC50 – there is no transcription of the abbreviation in the signature.

Response 2: Agree. We have, accordingly, modified

We have added the following abbreviations in the legend of the table 1:

“M: Male”

“F: Female”

We have added the following abbreviations in the legend of the table 2:

“MIC50: Minimun inhibitory concentration (MIC) that inhibits the growth of 50% of the tested microorganisms”

“MIC90: Minimun inhibitory concentration (MIC) that inhibits the growth of 90% of the tested microorganisms”

Comments 3: Line 21 – “...NDM-1 and OXA-23...” – requires a transcription of the abbreviation at the first mention in the text.

Response 3: We have agree with this comment.

We have modified the paragraph in the Introduction section as follows:

”Carbapenems-resistance is mainly due to the production of carbapenemases of the OXA-type hydrolyzing class D (class β-lactamases, CHDLs, oxacillinases), the chromosomal carbapenemase OXA-51 and acquired such as OXA-23, OXA-24/40 and OXA-58; and, less frequently, class B metallo-β-lactamases such as VIM (Verona Integron-encoded metallo-β-lactamase) and IMP (imipenemase metallo-β-lactamase ) [10] or NDM-1 (New Delhi metallo-β-lactamase) [11], which has recently been highlighted as a very worrying fact due to its possible gene expression without any fitness cost [12]. Co-production of OXA-23 and NDM-1 is infrequent, having been initially detected in African [13–18] and Asian [19–22] countries.

Comments 4: Line 18 - “In the summer of 2002…” – you write about the outbreak of the disease in 2002, but then you say that (line 23) “Patients with multidrug-resistant baumannii isolates (July 2022-May 2023)…” – only 20 years after the outbreak of the disease, have they started studying pathogenic multidrug-resistant A. baumannii isolates? And in the Discussion section, the authors describe an outbreak that occurred in 2016-2017. Please specify which event is in question in which year? And in what years were the samples taken?

Response 4: We agree with this comment. We have, accordingly, modified

It was a mistake. It has been conveniently explained and corrected in the comments of the first reviewer. Line 18 is not the year 2002, it is in the summer of the year 2022. Modified in the manuscript.

Comments 5:  Line 92 - “Presumptive Acinetobacter species were isolated on MacConkey agar…” – you need to add either a link or the composition of the medium.

Response 5: We agree with this comment. We have modified the paragraph in the Material and Methods section as follows:

“Presumptive Acinetobacter species were isolated on MacConkey agar and sheep blood agar (Becton Dickinson Microbiology Systems, Cockeysville, MD, USA) (https://www.bd.com/es-es/products-and-solutions/products/product-families/dehydrated-culture-media-and-additives)”

Comments 6: Line 91 – Section “2.2. Identification of Bacterial Isolates and Extraction of Carbapenemases Gene” – In the section, you should either add links to the listed methods with a brief description, or describe the techniques in detail

Response 6: The PCR assay was not developed in-house; accordingly, both the extraction protocol and PCR conditions were strictly followed in accordance with the manufacturer’s instructions.

For clarity we have included the following paragraph and added new references  as follows :

“Carbapenemase production was tested by a PCR assay (CarbaR+, Novodiag, Hologic), a platform for the multiplex qualitative detection of carbapenemases blaKPC, blaNDM, blaVIM, blaIMP, blaOXA-23, blaOXA-24, blaOXA-48/181, blaOXA-58 and colistin resistance gene mcr-1. Both, the extraction protocol and PCR conditions were strictly followed in accordance with the manufacturer’s instructions. The automated software interpretation of the sample results is based on the validation of internal extraction/inhibition control result. For sample processing, one overnight single colony grown on MacConkey agar plate was taken with a sterile loop and directly transferred in eNat preservation medium provided by the manufacturer. The colony was thoroughly suspended and the eNat tube vortexed for about 5 seconds and incubated for 30 minutes at room temperature for DNA release. After vortexing, 600 µL of the eNat suspension were added to the cartridge that was run on the Novodiag system according to the manufaturer´srecommendations (Mobidiag, Espoo, Finland) (Microb Drug Resist. 2021 Feb;27 (2):170-178) (J Microbiol Methods. 2021 Jan:180:105-106).”

Regarding to the identification of bacterial isolates

We have included the following paragraph as follows

“Bacterial identification was performed using a MALDI-TOF Biotyper instrument (Bruker Daltonics GmbH, Leipzig, Germany) by comparing the unique mass spectra of bacterial proteins with those in a database. All isolates were identified with a score higher than 2, indicating a more reliable identification, typically at the species level.”

Comments 7: It is not clear why the tables and figures are located at the end of the Results section, and not after the mention in the text during the presentation of the results.

Response 7: Agree.

We have modified the layout of the figures according to the suggestions.

Comments 8:  How many isolates were allocated in total for further work.

Response 8: All studies were performed on 24 representative isolates.

Comments 9:  There are no statistical data in the work.

Response 9: We agree with the reviewer that an association analysis would be valuable, however the small number of isolates per each ST will likely reduce the ability to detect true associations and may lead to a higher risk of errors. We did not perform any statistical analysis for this reason.

Comments 10:  Since the authors have the results of genome-wide sequencing, it would not be superfluous to add a genome-wide phylogenetic tree of the studied isolates.

Response 10: Regarding the characterization of the phylogenetic relationships among isolates to identify the strains involved in the described outbreaks, we consider that the analysis performed using cgMLST (based on 2,390 genes) provides sufficient resolution. This approach has been widely validated and employed in numerous studies for outbreak investigation and phylogenetic inference (references 2, 3, 4)

Ref 2. Hummel D, Juhasz J, Kamotsay K, Kristof K, Xavier BB, Koster S, Szabo D, Kocsis B. Genomic Investigation and Comparative Analysis of European High-Risk Clone of Acinetobacter baumannii  ST2. Microorganisms. 2024 Dec

2;12(12):2474.

Ref 3 Fernández-Cuenca F, Rodríguez-Pallares S, López-Cerero L, Gutiérrez-Fernández J, Bautista MF, Sánchez Gómez JA, Sánchez-Yebra Romera W, Delgado M, Recacha E, Pascual A. Regional distribution of carbapenemase-producing Acinetobacter baumannii isolates in southern Spain (Andalusia). Eur J Clin Microbiol Infect Dis. 2025 Feb 17

Ref 4 Ajoseh SO, Anjorin AA, Salami WO, Brangsch H, Neubauer H, Wareth G, Akinyemi KO. Comprehensive molecular epidemiology of Acinetobacter baumannii from diverse sources in Nigeria. BMC Microbiol. 2025 Mar 31;25(1):178.

Comments 11:  Please add the Conclusion section.

Response 11: We agree with this comment. We have included the following paragraph as follows :

“In conclusion, this study documents the clonal spread of an emerging NDM-1 and OXA-23 co-producing A. baumannii strain in Spain, belonging to the high-risk clone ST1 and linked to a Libyan patient. This highlights the risk of cross-border spread of multidrug-resistant microorganisms. Although infection control measures were successful in containing the outbreak,the integration of whole genome sequencing into CRAB surveillance in coordination with national reference laboratories was essential.

It is important to maintain surveillance strategies as the incidence of CRAB infections is expected to increase in the coming years.”

4. Response to Comments on the Quality of English Language

Point 1:

Response 1: We will send the manuscript for language review by a native specialist.

Round 2

Reviewer 2 Report

Comments and Suggestions for Authors

Dear authors!

Thank you for your work and for taking into account all the comments and correcting your manuscript. This demonstrates your responsibility and commitment to high-quality research. After the changes were made, the text became more structured and understandable. You have successfully eliminated ambiguities and improved argumentation, which makes your work more convincing.